# MET in Non-Small-Cell Lung Cancer (NSCLC): Cross ‘a Long and Winding Road’ Looking for a Target

**DOI:** 10.3390/cancers15194779

**Published:** 2023-09-28

**Authors:** Gianluca Spitaleri, Pamela Trillo Aliaga, Ilaria Attili, Ester Del Signore, Carla Corvaja, Chiara Corti, Jacopo Uliano, Antonio Passaro, Filippo de Marinis

**Affiliations:** 1Division of Thoracic Oncology, IEO, European Institute of Oncology, IRCCS, Via Ripamonti 435, 20141 Milan, Italy; pamela.trilloaliaga@ieo.it (P.T.A.); ilaria.attili@ieo.it (I.A.); ester.delsignore@ieo.it (E.D.S.); carla.corvaja@ieo.it (C.C.); filippo.demarinis@ieo.it (F.d.M.); 2Division of New Drugs and Early Drug Development for Innovative Therapies, European Institute of Oncology, IRCCS, 20141 Milan, Italy; chiara.corti@ieo.it (C.C.); jacopo.uliano@ieo.it (J.U.); 3Department of Oncology and Haematology (DIPO), University of Milan, 20122 Milan, Italy

**Keywords:** NSCLC, MET, MET exon 14 skipping mutations, MET amplification, MET overexpression, MET inhibitors, prognosis, immunotherapy, resistance mechanism

## Abstract

**Simple Summary:**

Around 3% of patients with Non-Small-Cell Lung Cancer (NSCLC) harbour a MET exon 14 skipping mutation (METex14). Early mutation identification is important for accurate treatment of these patients because they receive more benefit from chemotherapy than from immune checkpoint inhibitors (ICIs). Moreover, the treatment landscape of this disease has radically changed in recent years thanks to the introduction of new selective and potent MET inhibitors (MET-Is). The aim of our review was to summarize the historical milestones since the discovery of the MET pathway through studies investigating the role of MET in the prognosis of NSCLC patients harbouring MET alterations to the discovery of MET exon 14 skipping mutation, the real target of this pathway in NSCLC. Moreover, we focused on the results from pivotal clinical trials of MET inhibitors and on mechanisms of resistance to these drugs. The last section of this review is dedicated to future developments.

**Abstract:**

Non-Small-Cell Lung Cancer (NSCLC) can harbour different MET alterations, such as MET overexpression (MET OE), MET gene amplification (MET AMP), or MET gene mutations. Retrospective studies of surgical series of patients with MET-dysregulated NSCLC have shown worse clinical outcomes irrespective of the type of specific MET gene alteration. On the other hand, earlier attempts failed to identify the ‘druggable’ molecular gene driver until the discovery of MET exon 14 skipping mutations (METex14). METex14 are rare and amount to around 3% of all NSCLCs. Patients with METex14 NSCLC attain modest results when they are treated with immune checkpoint inhibitors (ICIs). New selective MET inhibitors (MET-Is) showed a long-lasting clinical benefit in patients with METex14 NSCLC and modest activity in patients with MET AMP NSCLC. Ongoing clinical trials are investigating new small molecule tyrosine kinase inhibitors, bispecific antibodies, or antibodies drug conjugate (ADCs). This review focuses on the prognostic role of MET, the summary of pivotal clinical trials of selective MET-Is with a focus on resistance mechanisms. The last section is addressed to future developments and challenges.

## 1. Introduction

MET (mesenchymal-epithelial transition) is a proto-oncogene situated on chromosome 7q21-q31 that encodes for transmembrane receptor tyrosine kinase normally expressed on epithelial cells. The Hepatocyte Growth Factor (HGF)/c-Met pathway is well characterized and recognized for its essential role in several cell vital processes in embryonic development, in the repair of injured tissues, as well as in carcinogenesis and tumour progression [1,2,3,4,5]. The HGF is produced by fibroblast cells, neutrophils and macrophages but was initially identified as a mitogen for hepatocytes [6,7]. Importantly, c-MET can interact with other ligands or cellular receptors, such as MET homolog RON (also known as MST1R), ROR1, CD44, integrins, and CD151, increasing the complexity of the pathway [8] (Figure 1). MET can be expressed in normal epithelial cells and overexpressed in several cancer cells, including Non-Small-Cell Lung Cancer (NSCLC) [9,10,11]. MET can play its role in cancer progression in different circumstances when cancer cells harbour MET dysregulation (MET addiction) [12,13,14]. MET alterations were involved in several cancers and consisted of MET mutations in renal papillary cancer (1–4%, first described in tumours), as well as MET amplification (MET AMP) in gastric cancer, melanoma (12%), de novo or as resistance to tyrosine kinase inhibitors in epidermal growth factor receptor (EGFR)-mutated NSCLC (15%), or colorectal cancer that develops resistance to EGFR antibodies [12,13,15,16,17].

Several landmark papers have demonstrated that NSCLC can harbour different MET alterations (MET+), such as MET overexpression (MET OE), MET phosphorylation (its activated form), MET amplification (MET AMP), or MET gene mutations and that these alterations can negatively affect the patients’ clinical outcomes [18,19,20,21,22,23,24,25,26,27,28,29,30]. On the other hand, several years had to pass before identifying the true therapeutic target among all these factors.

This review will focus on the first reports that have demonstrated that MET alterations are significant prognostic factors for NSCLC patients [18,19,20,21,22,23,24,25,26,27,28,29,30]. Then, we will describe the alterations of MET (mutations, amplification, and overexpression) and the methods of determination. Furthermore, we will extensively discuss the new selective MET inhibitors (MET-Is). An important section is dedicated to the role of immunotherapy in MET-dysregulated NSCLC. The last section will focus on the future developments.

## 2. Prognostic Role of MET in NSCLC

Some retrospective studies of surgical series have reported the prognostic role of MET dysregulation in NSCLC, which is summarized in Table 1 [18,19,20,21,22,23,24,25,26,27,28,29,30].

HGF and MET OE have been studied in four retrospective studies that found that these factors could be associated with aggressive disease phenotype (high ki-67 score), more extensive pathological disease (pT and pN), and worse overall survival (OS) [18,19,20,21].

Three independent studies have shown that MET mRNA levels were associated with more extensive disease (pN) and worse disease-free survival (DFS) [22,23,24].

Three studies reported that the MET gene copy number (GCN) in MET AMP, whatever the cut-off used, has an impact on survival [25,26,27]. Beau Fowler et al. showed that MET AMP (not stating the cut-off) was associated with a worse OS only in adenocarcinoma (not statistically significant) [25]. Okuda et al. showed that patients with stage II–IV and MET GCN > 3 copies had worse OS [26]. Cappuzzo et al. documented that patients with MET GCN ≥ 5 copies were associated with aggressive disease (higher grade score), more extensive disease (pT) and worse OS [27]. Instead, Park et al. failed to associate MET AMP (scored both through Colorado Criteria Cancer Center MET/CEP7 ratio ≥ 2 or Cappuzzo criteria GCN ≥ 5 copies) with clinical outcome, reporting that only MET OE was associated with worse OS [28].

Only two Asian studies included all three MET parameters, MET OE, MET AMP, and MET exon 14 skipping mutations (METex14), in their analyses, stating that METex14 are extremely rare (less than 5%) and can be associated with worse OS [29,30].

## 3. MET as Predictive Biomarker

### 3.1. MET Exon 14 Alterations

MET exon 14 encodes part of an important regulatory region and a binding site for cBL E3-ubiquitin ligase in the juxtamembrane domain of the MET receptor. Loss of this binding site leads to decreased ubiquitination, thereby reducing MET receptor internalization and degradation, resulting in increased MET levels [31,32,33]. MET ex14 alterations comprised point mutation, insertion, and deletion. Most frequent are disrupted splice sites that lead to MET ex 14 skipping in the RNA transcript and a truncated MET receptor without the Y1003 ubiquitin ligase binding site [34] (Figure 2).

Frampton et al. analyzed tumour samples from 38,028 patients and found 221 (0.6%) cases with METex14: 3.0% in lung adenocarcinoma and 2.3% in non-adenocarcinoma lung cancers [35]. Awad et al. reported the first retrospective study of METex14 NSCLC patients [36]. Out of 993 non-squamous NSCLC patients, they identified 28 (3%) METex14 NSCLC patients: most of them had adenocarcinoma histology, while 14% of them had pleiomorphic lung carcinoma [36]. In another retrospective series of 711 METex14 cases, 288 distinct METex14 mutations were identified [37]; of these, 88% were ever smokers, and 12% were nonsmokers. According to the histology, 478 (67.0%) were non-squamous, 79 (11.1%) squamous, and 24 (3.2%) adenosquamous. The most common METex14 mutations were D1028H (8.1%), D1028N (7.8%), c.3082 + 2T.C (5.0%), D1028Y (4.6%), and c.3082 + 1G.T (4.4%) [37].

An important issue is the diagnostic of METex14 mutations due to their relative rarity and complexity, as not all of them lead to an exon 14 deletion. Next-generation sequencing (NGS) is the best choice as a testing platform because it has the advantage of detecting other oncogenic drivers concurrently using one test performed on a single sample. Given the complexity of the MET exon 14 skipping mutations, different tools to target enrichment for NGS vary dramatically in their ability to detect these events [38]. DNA-based amplicon NGS assays have a low detection rate of METex14 (around 63% of the cases) [39]. The detection rate of METex14 can be improved by including fragment analysis and three additional amplicons to sequence exon 14 [40]. The hybrid capture library (which uses longer pieces of DNA region with respect to the primers) with optimized bioinformatics tools can improve the detection rate with respect to DNA-based NGS [41]. On the other hand, two reports have demonstrated that RNA-based NGS assays can be superior to DNA-based NGS tools in detecting METex14 [42,43]. However, one of the major caveats of RNA-based NGS is that RNA is more vulnerable to degradation than DNA, which leads to a reduction in the quality of RNA for formalin-fixed, paraffin-embedded samples [44]. An optimal solution could be represented by DNA-/RNA-based NGS assays such as Illumina or Thermo Fisher platforms [45,46]. Sun et al. theorized a calculation system to hypothesize which METex14 mutations really lead to exon 14 deletion and can be useful, above all, for extremely rare or novel mutations [47].

ESMO guidelines strongly recommend testing for METex14 skipping through an NGS panel. When using an RNA-based assay, it is mandatory that adequate internal validation and quality control measures are in place to ensure that rare gene mutations are not missing, and liquid biopsies can be used to test for molecular drivers. However, all patients with a negative blood test still require tissue biopsy [48].

### 3.2. MET Amplification

The MET amplification (MET AMP) is due to an increase in the copy number of the MET gene (GCN), resulting in protein overexpression. True gene amplification results from gene duplication should not be confused with increased MET copy number due to high polysomy where there is complete chromosome duplication leading to multiple copies of chromosome 7 in tumour cells. Distinguishing between these two mechanisms is critical as true MET amplification is thought to drive oncogenesis, whereas polysomy usually does not [33] (Figure 2). 

MET amplification (MET AMP) as a ‘druggable driver’ has a longer and more complex history because, in this circumstance, we need to select the real gene-addicted tumours to avoid the ‘dilution effect’ that consists of including a heterogeneity patient population that comprises polysomy and low gene amplification.

In fact, the frequency of MET GCN in NSCLC ranges from 0.7% to 21%, depending on the technique used and the cut-off point for positivity [49]. Fluorescence in situ hybridization (FISH) was the first validated method for detecting high MET copy numbers and, to date, remains the gold standard method [48].

One of the first cut-offs used was the University of Colorado Cancer Center (CCC) criteria, which established the gene amplification on the basis of GCNs and the ratio between GCN and centromere of chromosome 7 (MET/CEP7). It also distinguishes low polysomy (≥4 copies in 10–40% of cells), high polysomy (≥4 copies in ≥40% of cells), and gene amplification (presence of tight gene clusters and a MET/CEP7 ratio of ≥2 or 15≥ copies of MET per cell in ≥10% of cells) [50]. The next cut-off was established by Cappuzzo et al., where MET GCN was classified into those with a mean greater than or equal to five copies per cell versus less than five copies. Using this system, they identified 48 (11.1%) patients out of 435 cases, where true amplification was identified in 18 (4.1%) patients [27]. Subsequently, two studies used the Cappuzzo criteria and CCC criteria, finding MET+ cases in 7% and 4.2%, while the true amplification was only 2.4% and 1.1% [28,30].

Noonan et al. analyzed 1164 patients with adenocarcinoma with information on MET/CEP7 ratio and 686 with information on mean MET per cell value from two separate cohorts, and the GCN was reported by two methods: mean MET copy number per cell value (low ≥ 5 to <6 copies; intermediate ≥ 6 to <7 copies; high ≥ 7 copies) and the ratio MET/CEP7 (low ≥ 1.8 to ≤2.2; intermediate > 2.2 to <5; or high ≥ 5) [51]. Out of 686 cases, 99 (14%) had a MET ≥ 5 copies (FISH+), and 52 of 1164 (4.5%) had a MET/CEP7 ratio ≥ 1.8, meanwhile only 4 (0.03%) patients had a high amplification (ratio ≥ 5). Overall, 61% of 1164 patients had concomitant gene mutations. According to the MET/CEP7 ratio, the concomitant gene mutations were distributed as the following: low (15 of 29, 52%), intermediate (9 of 18, 50%), and high (0 of 4, 0%, *p* = 0.04). This rigorous cut-off (ratio > 5) helped to identify extremely rare true amplificated cases (less than 1%) without other concomitant gene alterations [51]. However, this analysis was not conclusive, and even now, several cut-offs are being used.

It should be noted that MET AMP has been described in patients with NSCLC not previously exposed to tyrosine kinase inhibitors (TKIs) (de novo MET amplification) or as a mechanism of resistance to TKIs, mainly to EGFR (acquired MET amplification).

### 3.3. MET Overexpression

MET protein overexpression ranges from 15 to 70% in NSCLC [28]. The MET receptor overexpression (MET OE) is measured by immunohistochemistry (IHC). It is found that there is a poor correlation with MET gene amplification or mutation. In the Lung Cancer Mutation Consortium multi-institutional cohort study of seventy-one IHC-positive cases, only one (1%) was MET AMP, and two (3%) were METex14-mutated; instead of the 110 MET IHC-negative cases, two (2%) were MET-amplified [52]. In addition, several phase 3 trials with MET overexpression treated with MET-targeted therapies found that protein overexpression was an unreliable biomarker; thus, the utility of this biomarker is limited in clinical practice [52].

## 4. Targeting MET

In the past, two randomized phase III trials with onartuzumab (a monoclonal antibody) and tivantinib (a selective MET TKI inhibitor) in the second/third line were conducted in an unselected NSCLC patient population, and these trials failed to meet their primary endpoint in improving the OS [53,54] (Appendix A). 

Additionally, several MET-Is have been developed despite the failure of these first approaches [53,54]. MET-Is can be distinguished because of the mechanism of trapping MET: type I inhibitors compete with ATP to the ATP-binding pocket of the active conformation of MET, type Ia (crizotinib) interacts through the G1163 site, while type Ib (capmatinib, tepotinib, and savolitinib) interaction is independent of the G1163 site; type II ATP-competitive MET kinase inhibitors (cabozantinib, glesatinib, and merestinib) are defined by their ability to inhibit MET in its inactive state [55,56].

### 4.1. Crizotinib

Crizotinib is a type Ia MET-I through G1163 residue interaction and inhibits the ATP-binding pocket of active kinase [57]. Appendix A summarizes all the cases reported of crizotinib in patients with NSCLC harbouring MET alterations [36,58,59,60,61,62,63,64,65].

The AcSé was a French phase II trial in heavily pretreated patients with MET+ or ROS1+ NSCLC [66]. They screened more than 5000 patients: 252 out of 4193 evaluable patients (6%) resulted positive for MET AMP (cut-off GCN ≥ 6 copies), and 74 out of 1192 (6.2%) were positive for all MET mutations, including METex14. Results among the 28 patients with all MET mutations were ORR 36%, mPFS 2.4 months, and OS 8.1 months. In the 25 patients with METex14, the ORR was 40%, the mPFS was 3.6 months, and the OS was 9.5 months [66]. For the 25 patients with MET AMP, the ORR was 32%, and the responses according to the GCN level were high GCN 0/1 (0%), intermediate GCN 5/6 (83%), low GCN, or polysomy 2/15 (13%). The mPFS and mOS were 3.2 and 7.7 months, respectively [66].

The METROS was an Italian multi-center phase II trial of patients with MET+ or ROS1+ NSCLC [67]. They screened 505 patients, and a total of 37/433 (8.5%) were MET dysregulated; of them, 26 patients were enrolled: 10 had METex14 and 16 had MET AMP (cut-off MET/CEP7 ≥ 2). The ORR was 27%, the mPFS was 4.4 months, and the OS was 5.4 months. Among the 10 patients with METex14 skipping mutations, they recorded two partial responses (PRs) (20%). No intra-cranial responses were noted in the MET cohort of the trial [67]. Among the 16 patients with MET AMP, the responses according to the GCN were high level 0/2 (0%) and intermediate level 5/14 (36%) [67].

The expansion cohort of the phase I trial PROFILE-1001 included 69 patients with METex14 who had previously been treated for NSCLC [68]. Among the 65 evaluable patients, the ORR was 32%, the median duration of response (mDoR) was 9.1 months, the mPFS was 7.3 months, and the OS was 20.5 months. Importantly, they did not record any difference in ORR among type or localization of MET mutations. They included only two patients with concomitant MET AMP and observed a durable response exceeding 6 months. They reported the following concomitant gene mutations: p53 (38%), MDM2 (20%), and CDKN2A (20%); however, they reported no clinical efficacy data among patients with concomitant gene mutations. They observed that patients with negative circulating tumour DNA (ctDNA) did not differ in ORR with respect to patients with positive ctDNA. However, these patients were associated with longer mDoR (7.8 vs. 3.6 months, respectively). The most common adverse events (AEs) were oedema (51%, G ≥ 3 = 1%), vision disorder (45%, G ≥ 3 = 0%), nausea (41%, G ≥ 3 = 0%), diarrhea (39%, G ≥ 3 = 0%), and vomiting (29%, G ≥ 3 = 0%). The G3/4 AE rate was 29%, leading to dose reduction (38%) and drug discontinuation (7%). Of note, one patient died due to a grade 5 interstitial lung disease (ILD) [68].

As for the cohort of 38 pretreated patients with MET AMP (cut-off MET/CEP7 ratio ≥ 1.8) NSCLC in the PROFILE-1001 study, 21 pts had high, 14 pts had medium, and 3 pts had low MET AMP [69]. Concurrent METex14 was detected in 2 of 17 (11.8%) patients (1 each in the high and medium groups). The overall ORR was 28.9%, and the responses were higher according to the amplification level: high group (38.1%, with two complete responses [CR]), medium group (14.3%), and low group (1/3), respectively. The mDoR was 5.2 months, and the mPFS was 5.1 months. Again, the mPFS was higher in the high MET AMP group (6.7 months) with respect to the medium (1.9 months) or the low group (1.8 months). Overall, the OS was 11 months. Importantly, they observed a high concordance between FISH and NGS analysis; however, patients selected by NGS had a higher ORR (40% 6/15), and in patients with NGS positive for MET and without other concomitant gene mutations, the ORR was 54.5% [69].

Song et al. reported a retrospective series of 15 patients with MET AMP NSCLC (cut-off MET/CEP7 ≥ 1.8) collected from five Chinese hospitals from 2014 to 2016 and treated with crizotinib [70]. They reported 73.3% PRs and 20% of stable disease (SD). The mPFS was 6.5 months, and the OS was 31 months. In this report, patients with high MET AMP were associated with a longer mPFS than patients with intermediate MET AMP (8.6 months vs. 4.4 months, respectively) [70]. 

### 4.2. Capmatinib

Capmatinib is a type Ib selective MET-I. Preclinical data have shown that capmatinib is active in MET+ cancer models both in vitro and in vivo [71,72].

The phase I trial enrolled 55 patients with MET+ NSCLC (MET IHC 2+ or 3+/MET GCN ≥ 6 or MET/CEP7 ratio ≥ 2) treated with capmatinib and attained an ORR of 20%, 47%, and 100% among all the patients, 15 patients with MET GCN ≥ 6 and 4 patients with METex14, respectively [73].

GEOMETRY-mono 1 was an international, multi-center, phase II trial in 364 patients with MET+ NSCLC treated with capmatinib and distributed in several cohorts based on MET-type alteration and prior treatments [74]. In this trial, METex14 or MET AMP was centrally assessed. Also, neurologically stable brain metastases were allowed. In this study, 151 patients with METex14 were enrolled: 28 (cohort 5a) plus 32 (cohort 7 without fasting restrictions) treatment-naïve patients; 31 patients who received capmatinib as second-line treatment (cohort 6); and 69 patients who received capmatinib as ≥2-line treatment (cohort 4) [74,75]. Among the 69 pretreated patients (cohort 4), the ORR was 41%, the mDoR was 9.7 months, and the mPFS was 5.4 months. For patients enrolled in cohort 6 (second-line treatment), the ORR was 48%, the mDoR was 6.9 months, and the mPFS was 12.4 months. Importantly, among the 28 treatment-naïve patients (cohort 5a), the ORR was 68%, the mDoR was 12.6 months, the mPFS 12.4 months, and the OS was 20.8 months. Among the confirmatory cohort 7 (32 patients), the ORR was 65.6% and the mPFS 10.8 months [75]. Of note, no significant differences have been recorded among various types of MET mutations nor concomitant MET AMP [74]. As previously demonstrated, capmatinib can cross the blood–brain barrier (BBB); in a post hoc analysis of 13 evaluable patients out of a total of 14 patients with baseline brain metastases, they observed seven intracranial responses (four naïve to brain radiotherapy), and four of them were CR [74,76,77]. Another post hoc analysis showed that among the 30 patients who were treated with prior immunotherapy (ICI) compared with 57 patients who were not pretreated with ICI, the activity was 62.5% vs. 33.8%, mDoR 10 vs. 7 months, mPFS 8.3 vs. 5.3 months, respectively [78].

GEOMETRY mono-1 also included patients with MET AMP NSCLC regardless of GCN cut-off [74]. All cohorts had negative results. Cohort 1b (42 patients, GCN 6–9), cohort 2 (54 patients, GCN 4–5), and cohort 3 (30 patients, GCN < 4) were prematurely closed due to futility with an ORR of 15%, 12%, 9%, and 7%, respectively. In cohort 1a (69 pretreated patients, GCN > 10), the ORR was 29% (a negative result because it was lower than the boundary fixed when the trial was designed, which was 35%), the mDoR was 8.3 months, and the mPFS was 4.2 months. Also, in cohort 5a (15 naïve patients, GCN > 10), the ORR was 40% (a negative result because it was lower than the boundary of 55%), the mDoR was 7.5, and the mPFS was 4.1 months [74].

The median duration of exposure to capmatinib varied across the cohorts, ranging from 6.6 weeks to 48.2 weeks, and the most common AEs were oedema 51% (G ≥ 3= 9%), nausea 45% (2%), vomiting 28% (2%), and creatinine 24% (0%). Of note, the ILD was 4.5% of them, and one patient died. The G3/4 rate was 67%, AEs leading to dose reductions were 23%, and drug discontinuation was 11%. Importantly, a lower incidence of gastrointestinal AEs was observed in cohorts 6 and 7 (capmatinib administered without fasting restrictions) [74].

### 4.3. Tepotinib

Tepotinib is a highly selective type of Ib MET-I [79]. A phase I trial and three phase I/II trials have shown that tepotinib is safely administered and has promising activity in patients with MET+ hepatocarcinoma and EGFR+ NSCLC progressing to TKIs and that developed MET OE or AMP [80,81,82,83].

VISION was an international, multi-center, multicohort, phase 2 trial to evaluate the efficacy and safety profile of tepotinib in 337 MET+ NSCLC patients [84]. This trial included advanced NSCLC harbouring METex14 or MET AMP (centrally assessed on either tissue or liquid biopsy), and neurologically stable brain metastases were allowed.

A total of 313 patients with METex14 NSCLC were enrolled: 152 (cohort A) and 161 (confirmatory cohort C). Patients with brain metastases were 20% in cohort A and 15% in cohort C [85,86,87]. There were 164 treatment-naïve patients (69 in cohort A plus 95 in cohort C) and 149 pretreated patients (83 in cohort A plus 66 in cohort C). Overall, the median follow-up was 32.6 (0.3–71.9) months (cohort A 35 months and cohort C 18 months). Overall (313 patients), the ORR was 51.4%, the mDoR was 18 months, the mPFS was 11.2 months, and OS was 19.6 months [87]. Among pretreated patients (149), ORR was 45.0%, mDoR was 12.6 months, and OS was 19 months. In the treatment-naïve patients (169), the ORR was 57.3%, the mDoR was 46.4 months, the mPFS were 10.3 (cohort A) and 15.9 (cohort C) months, and OS was 19.1 (cohort A) and 21.1 months (cohort C) [86,87]. Results were more robust in the tissue-biopsy group both for cohorts A and C [84,86,87]. In cohort C, outcomes in treatment-naive patients with T-positive METex14 (n = 69) were further improved, with an ORR of 65.2%, mDoR not reached, mPFS of 16.5 months, and OS of 28.5 months [87]. They did not observe any significant differences in the patients regarding response to treatment among various types of MET mutations nor concomitant MET AMP [84]. Moreover, 51 patients were evaluated for molecular response. Of these patients, 34 had a molecular response, and this correlated with the radiological response: 84% of them had a disease control rate (DCR), and 71% had ORR [84]. A post hoc analysis of prior treatments in cohort A identified that 74 patients had received prior platinum-based chemotherapy, 29 patients had received ICI, and 10 pts had received both. The efficacy of tepotinib was similar, irrespective of the type of prior treatments [88]. Another post hoc analysis based on RANO criteria showed that among 57 patients with brain metastases at baseline, the DCR was 88% with an intra-cranial mPFS of 20.9 months, and the intracranial ORR was 66.7% with an intracranial mDoR not reached among the 15 patients with intracranial target lesions [86,87,89].

Twenty-four patients with NSCLC harbouring MET AMP have been enrolled in cohort B of the VISION trial [90]. The MET AMP was detected by a liquid biopsy assay (73-gene GUARDANT360) with a cut-off of GCN ≥ 2.5. They screened 3205 patients and found 70 patients (2.2%) positive for this cut-off. Of them, 24 patients were enrolled in cohort B and were treated with tepotinib. Seven patients (29%) were treatment-naïve, and seventeen patients (71%) had been previously pretreated. Among all the patients, the ORR was 41.7%, the mDoR was 14.3 months, the mPFS was 4.2 months, and the OS was 7.5 months. Among the 17 pretreated patients, the ORR was 29.5% (5/17 patients), and the mPFS among the 11 patients (treated in second-line setting) and 6 patients (third-line setting) were 13.6 and 1.7 months, respectively. Among the seven treatment-naïve patients, the ORR was 71.4% with an mPFS of 15.6 months. Regarding the biomarker analysis, they found that the patients that attained better clinical outcomes were those with focal amplification (14 patients with ORR 56% and mPFS of 15.6 months), those with RB1 wild-type (19 patients with ORR 52%, but with an mPFS of 4.5 months), those with MYC diploidy (18 patients, ORR 55% with an mPFS of 13.6 months), those with low ctDNA burden (12 patients, ORR 66%, and an mPFS not estimable), and the 14 patients who achieved molecular response (ORR 71% and mPFS 13.6 months) [90].

The most common AEs with tepotinib were oedema 66.5% (G ≥ 3, 10.9%), nausea 23.3% (0.6%), hypoalbuminemia 23% (3.2%), diarrhea 22.4% (0.3%), and creatinine level increase 21.7% (0.6%). The TRAEs G3-4 was 34.8%, and the AEs leading to dose interruptions, dose modifications, or drug discontinuation were 42.5%, 33.5%, and 14.7%, respectively [85,86,87].

### 4.4. Savolitinib

Savolitinib is a type Ib MET-I. Preclinical studies have demonstrated its activity towards MET AMP gastric cancer patient-derived tumour xenograft models [91,92].

A single-arm phase 2 study of savolitinib in 70 patients with METex14 NSCLC pretreated with one or more standard treatments or unfit for standard treatment across 32 hospitals in China [93]. This trial enrolled 25 (36%) patients with sarcomatoid histologies, and 15 (21%) had brain metastases. The median follow-up was 17.6 months. The ORR was 49.2% in the efficacy patient population (61 patients) and 42.9% in 70 intention-to-treat (ITT) patients, the mDoR was 8.3 months, the mPFS was 6.8 months, and the OS was 12.5 months. Responses were irrespective of the type or location of MET mutations, and eight patients with concomitant MET AMP had better ORR and longer mPFS. Among the 25 patients with sarcomatoid histologies, they attained an ORR of 40%, a mDoR of 17.9, and an mPFS of 5.5 months. Among 45 patients with other histologies (mostly adenocarcinoma), the ORR was 45%, and the mDoR and the mPFS were 8.3 and 6.9 months, respectively. Among the 42 previously pretreated patients, the ORR was 40.5%, the mDoR and the mPFS were 9.7 and 6.9 months, respectively. Among the 28 treatment-naïve patients, the ORR was 46.4%, and the mDoR and the mPFS were 5.6 and 5.6 months, respectively. In a post hoc analysis among the 15 patients with brain lesions at baseline, the global ORR was 46.7%, and the DCR was 93% with a global mPFS of 6.9 months. The only three patients with brain target lesions attained an intracranial PR. Among 65 evaluable patients with sufficient tumour samples for NGS, the most frequent concomitant gene mutations were p53 (49%), MDM2 (25%), and POT1 (in 8 pts with sarcomatoid histologies). Patients with concomitant p53 mutations had significantly less ORR and showed a trend for a shorter mPFS; among the 8 patients with sarcomatoid histologies harbouring POT-1 mutations, they showed a non-significant lower ORR and shorter mPFS with respect to patients with POT-1 wild type [94]. The most common AEs with savolitinib were peripheral oedema (any grade 56%, G ≥ 3 =9%), nausea (53%, G ≥ 3= 0%), hypoalbuminemia (41%, G ≥ 3 = 1%), increased alanine aminotransferase (39%, G ≥ 3 = 13%), and aspartate aminotransferase (39%, G ≥ 3 = 10%). The treatment-related G3/4 toxicity rate was 46%. AEs leading to dose reductions were 44%, and the AEs leading to drug discontinuation were 20% [93].

Currently, the clinical efficacy of savolitinib is being investigated in several tumours and EGFR+ NSCLC that progress to osimertinib and develop MET OE/MET AMP [94].

Table 2 summarizes the results from phase II trials of selective MET-Is in patients with METex14 or MET AMP NSCLC, and Appendix A depicts the details from phase II trials of capmatinib and tepotinib.

## 5. Resistance Mechanisms to MET-Is

### 5.1. Intrinsic Resistance

It refers to primary resistance and, to our knowledge, is a phenomenon poorly described for MET+ NSCLC treated with MET-I. Moreover, there are currently no translational studies that have investigated epigenetic alterations that could have helped us better understand the mechanisms of resistance to inhibitory METs. Type I MET inhibitors’ activity does not seem to be negatively influenced by the different types or localizations of the METex14 mutations [68,74,84,93]. Subgroup analyses for evaluable patients with sufficient tumour samples for NGS from PROFILE-1001 and phase II trial of savolitinib and NGS analyses from liquid biopsies from the VISION trial showed that patients with METex14 NSCLC could have concomitant gene mutations and that the most common are p53 (38–49%), MDM2 (20–25%), and CDKN2A (20%) [68,84,93]. Moreover, post hoc analyses showed that p53 could negatively affect the response (savolitinib trial), and a trend has been documented for the lasting benefit of p53 (VISION and savolitinib trials) and POT-1 in few patients with sarcomatoid histologies (savolitinib trial) [84,93,95]. Importantly, for patients with high MET AMP NSCLC, the absence of concomitant gene mutations was associated with higher ORR in PROFILE-1001 [69].

### 5.2. Acquired Resistance

In METex14 or MET AMP NSCLC, patients treated with MET-Is acquired mutations in the tyrosine kinase domain (TKD) and can sustain resistance to these inhibitors. Importantly, D1228 and Y1230 seem to mediate resistance to type I inhibitors by disrupting drug binding, while L1195 and F1200 mutations seem to confer resistance to type II inhibitors [96,97,98,99,100,101,102,103,104]. Moreover, an off-target mechanism of resistance to MET inhibitors involving KRAS gene amplification or mutations has been proved both in preclinical and clinical models [105,106]. One preclinical study with selected MET-Is showed that mutations within the MET activation loop (D1228N, Y1230C/H) were associated with resistance to type I MET-Is but remained sensitive to type II inhibitors (glesatinib) [98].

In a retrospective study, Recondo et al. reported 20 patients with METex14 NSCLC treated with MET-Is [102]. They found genomic alterations (GAs) potentially related to resistance to MET-Is in 15 (75%) patients: 7 (35%) patients developed single or compound MET mutations on TKD in codons G1163R, D1228H/N, Y1230C/H/S, and L1195V (associated with type I inhibitors crizotinib and capmatinib) and H1094Y and L1195V (type II inhibitor glesatinib) plus high MET AMP, while 9 (45%) patients were associated with off-target alteration including KRAS (mutations or amplifications) and gene amplification of EGFR, HER3, and BRAF; one patient developed both on- and off-target mechanisms of resistance. It should be noted that two out of six patients (33%) were sensitive to the MET-I switch (the first case was initially treated with crizotinib and subsequently with merestinib; the second was treated with glesatinib and then with crizotinib) [102].

A second Chinese retrospective study has been conducted with 86 patients with NSCLC harbouring MET TKD mutations [103]. Importantly, they screened three different cohorts of patients (treatment-naïve patients with EGFR+ NSCLC treated with either EGFRTKI and/or MET inhibitors), and they found that ‘acquired’ MET mutations are extremely rare in treatment-naïve patients (0.06%) and have higher frequency after EGFR/MET-Is [103].

From a small phase II trial of capmatinib in 20 patients with METex14 NSCLC after crizotinib treatment, among the sixteen patients with detectable ctDNA, five (31%) had acquired MET mutations, three (19%) had MAPK pathway alterations, and two (13%) had ERBB pathway alterations [104]. From the VISION trial of tepotinib, 52 pts with progression had end-of-treatment liquid biopsy samples [95]. Emerging MET resistance mutations (Y1230H/C and D1228H/N, plus three unknown functions G685E, G344R, and S156L mutations) occurred in seven (13%) patients and off-target mechanisms p53/RB1 mutations (6/35), EGFR/HER2 amplifications (4/35), and PI3K/RAS mutations (3/35) [95]. Two independent studies have shown that KRAS alterations (amplification or mutations) can play a significant role in acquired resistance to MET inhibitors [105,106].

As for MET AMP NSCLC, nine evaluable patients out of twenty-four patients enrolled in cohort B (MET amplification) of the VISION trial have had liquid biopsy profiles at disease progression: two (22.2%) of them had acquired MET resistance mutations; one patient had D1228H/N/Y, Y1230C/H, and D1231N; and another had D1228N/H, Y1230H, and D1231N (90). Figure 3 and Table 3 summarize the intrinsic and acquired mechanisms of resistance to MET-Is.

## 6. MET and Immunotherapy

Pivotal clinical trials of ICI have excluded or not included patients with known MET alterations NSCLC; therefore, our knowledge of efficacy in this target patient population can only be derived from translational research and retrospective trials [107]. MET OE NSCLCs were associated with higher PD-L1 expression and T-cell infiltration [108].

MET AMP NSCLC correlated with higher PD-L1 score, CD8+ T-cell infiltration, higher incidence of concomitant gene mutations and worse OS [109]. Patients with high MET AMP NSCLC (stated as >10 copies) experienced a statistically significant benefit on OS when treated with ICI added to chemotherapy [110].

Instead, there are conflicting results regarding patients with METex14 NSCLC and PD-L1 expression: one study showed that they are associated with higher PD-L1 score [111], another study documented higher CD8+ T-cell infiltration but no higher PD-L1 levels [112], while another third study found that they did not have high median Tumour mutational burden (mTMB) nor high PD-L1 levels [113].

Table 4 summarizes these retrospective studies of second or further lines of ICI in MET+ NSCLC.

Sabari et al. reported 24 patients with METex14 NSCLC treated with ICI; the ORR was 17%, and the mPFS was 1.9 months. Responses were not enriched in tumours with PD-L1 ≥ 50% nor high TMB [113].

Dudnik et al. reported 14 patients with METex14 NSCLC and 5 patients with MET AMP (cut-off not specified) NSCLC [114]. The activity of ICI was modest in both groups of patients: in the first, the ORR was 12% (none out of two patients responded to treatment among those with PD-L1 ≥ 50%) with an mPFS of 4 months (1.9 months in the two patients with PD-L1 ≥ 50%); in the latter, the ORR was 25% (one out of four patients included in the analysis) with an mPFS of 4.9 months [114].

The IMMUNOTARGET registry was a large European multi-center retrospective trial also involving EGFR and KRAS and enrolled 551 patients with gene-addicted NSCLC [115]. ICI was given mostly as the second- or third-line setting. Thirty-six patients had MET+ NSCLC (twenty-three patients with METex14 plus thirteen with MET AMP, cut-off not indicated), the median PD-L1 score was 30, and 76% of the patients were ever smokers. Among these patients, the ORR was 16%, the mPFS (principal endpoint of the study) was 3.4 months, and the OS was 18.4 months. The mPFS was not augmented, regardless of whether we considered the MET alteration subtype, the smoking status, or PD-L1 expression [115].

The GFPC01-2018 study has enrolled 30 patients with METex14 NSCLC (enriched for PDL-1 positive and ever smokers) treated with nivolumab or pembrolizumab and showed an ORR of 36%, an mPFS of 4.9 months, and an OS of 13.4 months [116].

This was recently reported in two retrospective studies. In the first study of 87 patients with METex14, NSCLC in the mPFS of ICI-based regimens was numerically low (2.4 months in patients with high PD-L1) [117]. In the second study of 248 patients with METex14 NSCLC, the mPFS of ICI was inferior with respect to chemotherapy in the first-line setting (229 patients, 3.6 vs. 5 months) and second-line setting (158 patients, 3.3 vs. 3.9 months) [118].

Post hoc analyses from GEOMETRY-mono1 and VISION trials have demonstrated that the activity of the MET inhibitor (capmatinib or tepotinib) was irrespective of prior therapies (including ICI); however, the median time to treatment of ICI in the VISION trial was less than 4.5 months, and the OS of patients treated with ICI alone was inferior respect to patients treated with chemotherapy +/− ICI (15.8 vs. 20 months) [78,119].

## 7. Future Developments

Currently, there are new agents targeting MET that are at an advanced stage of clinical research (completed phase I/II) and are summarized in Table 5.

Amivantamab is a new bifunctional monoclonal antibody against EFGR and MET [120,121]. Chrysalis is a phase I clinical trial in which 55 patients with METex14 have been enrolled (cohort METex14); 18 were treatment-naïve, and 28 were pretreated. The ORR was 33%, the mDoR was not estimable, and the mPFS was 6.7 months. The activities according to prior treatment were the following: ORR of 57% vs. 45% vs. 17% and mPFS NR vs. 8.3 months vs. 4.2 months in treatment-naïve, MET-I-naïve, and MET-I-pretreated patients, respectively [122]. Moreover, amivantamab has already been approved by the FDA and EMA for EGFR exon 20 insertion mutations NSCLC, and it is currently in clinical research for patients with EGFR+ NSCLC progressing on TKIs (MARIPOSA-2 and PALOMA-3) [123,124].

Telisotuzumab vedotin (ABBV-399; Teliso-V) is a cleavable antibody–drug conjugate (ADC) of a recombinant MET–targeting humanized monoclonal antibody (ABT-700) and monomethyl auristatin E (MMAE), a potent inhibitor of microtubule polymerization [125]. The phase I trial in 48 patients with MET OE solid tumours showed that the drug is well tolerated and has encouraging results [126]. A phase Ib of Teliso-V plus erlotinib has been carried out in 42 patients with NSCLC; most of them (28) were EGFR+ that were progressing to EGFRTKI [127]. Luminosity was a phase 2 trial in 136 MET OE (stated as high, ≥50% 3+, and intermediate, 25 to <50% 3+, in non-squamous cohort as ≥75% 1+ for squamous cohort) pretreated (second- or third-line) patients [128]. ORR was 36.5% with a mDoR of 6.9 months in the non-squamous (NSQ) EGFR wild-type (WT) cohort (52.2% in the c-Met high group and 24.1% in the c-Met intermediate group), but it was modest in the NSQ EGFR mutant and SQ cohorts [128,129]. Therefore, a phase II trial of teliso-V in 70 patients with MET AMP NSCLC patients is ongoing (NCT05513703) [130].

TPX-002, or elzovantinib, is a new type I TKI directed against MET, SRC, and CSF1R. SHIELD-1 TRIAL is an ongoing phase I trial investigating the safety and activity of elzovantinib in 52 patients with advanced solid tumours harbouring MET alterations. Fifty-two patients have been enrolled across seven dose levels, including thirty NSCLC patients (twenty with METex14, eight with MET AMP, and two with other MET mutations). Among 11 TKI-naïve NSCLC patients, the ORR was 36% among all the doses (43% among the 7 patients treated at the RP2D) [131].

Sym015 is a mixture of two humanized antibodies and triggers MET degradation [132]. The Sym015-01 was a phase I/II trial of Sym015 [133,134]. Forty-five patients have been enrolled in the second part of the trial [134]. Of 20 NSCLC patients, 5 had confirmed PR (ORR 25%; 2/8 MET AMP and 3/12 METex14); among the 10 TKI-naïve NSCLC patients (7 MET AMP and 3 METex14), the ORR was 50%; among the 10 TKI-pretreated NSCLC pts (10 METEx14 of whom one with additional MET AMP), there was a DCR of 60%. Overall, the mPFS was 5.5 months: 6.5 and 5.4 months for MET-TKI-naïve and pretreated, respectively [134].

REGN5093 is a new biparatopic MET antibody in which each arm of the antibody recognizes a distinct epitope of MET, which induces MET internalization and degradation. The phase I study enrolled in the dose expansion cohort 44 patients with METex14, MET AMP (GCN ≥ 6 or MET/Chr 7 ratio ≥ 4, or MET gene fold change ≥ 2), or OE (3+ or H score ≥ 200) [135]. In cohort 1A (11 patients with MET TKI-pretreated METex14 NSCLC), there were no responses; in cohort 1B (10 patients with MET TKI-naïve METex14), there was an ORR of 30%. Cohorts 2A, 2B, and 2C were catered for MET AMP or MET OE EGFR WT or EGFR+ NSCLC, attaining an ORR of 12.5% [135]. Preclinical data have shown the activity of a new ADC of REGN5093 conjugated to a novel maytansinoid M114 in EGFR+ NSCLC cancer cells refractory to EGFR-TKI [136].

In the early-stage setting, a perioperative usage of capmatinib is investigated in the GEOMETRY-N (NCT04926831) and the neoadjuvant chemo-immunotherapy (toripalimab plus chemotherapy) is explored in 30 patients with rare molecular driver NSCLC (NCT05800340).

In the advanced setting, MOMENT (NCT05376891) is a registry of patients with METex14 NSCLC on active treatment. NCT05567055 and NCT04739358 are clinical trials of capmatinib and tepotinib in 35 and 65 patients with active brain metastases, respectively.

As far as the new drugs against MET are concerned, they are in different stages of clinical research; this is summarized in Appendix A [137]. From a structural and functional point of view, we can distinguish them in MET-Is (including bozitinib, ningetinib, elzovantinib, BPI-9016M, glumetinib, ABN401, and DO-2) [138,139,140]; bispecific or trispecific antibodies (MCLA-129, EMB-01, CKD-702, and GB263T) [141,142,143]; new biparatopic topic monoclonal antibodies (Sym015); and ADCs (ABBV400 or telisotuzumab or teliso-topoismoerase I inhibitor, ABBV-399 or teliso-V, REGN5093-M114, and MYTX-011) [144,145].

**Table 5 cancers-15-04779-t005:** Results of phase I/II trials of new anti-MET drugs.

Trial/Ref	Drug Class	Drug & Schedule	Pt N	ORR	mPFS	AEs G3/4 R	Most Common AEs % (≥3%)
CHRYSALIS Krebs 2022 [138] NCT02609776	bsAb	Amivantamab Ab anti-EGFR/MET Fortnightly 1400 (>80 kg) 1050 (≤80 kg)	46 METex14 7 tx-naïve 15 MET-I-naïve 24 pre-MET-I	15/46 (33%) 4/7 (57%) 7/15 (45%) 4/24 (17%)	6.7 mos NE 8.3 mos 4.2 mos	24% Int. 21% Mod. 12% Disc. 5%	IRR 69 (5) SR 31 (2) DA 40 (0) Par. 38 (0) Fat. 31 (4) Alb. 27 (2)
Luminosity Camidge 2022 [128,129] NCT03539536	ADC	Teliso-V Ab antiMET-vedotin 2.7 mg/kg q3w	136 MET IHC+ 28 Sq 44 Non-Sq 44 EGFR+	3/27 (11.1%) 19/52 (36.5%) 5/43 (11.6%)	NR NR NR	48% Int. NR Mod. NR Disc. 13%	NP 25 (4) N 22 (1) Alb. 20 (1)
SHIELD-1 Hong 2021 [131] NCT03993873	MET-I	Elzovantinib (TPX-022) MET/SRC/CSF1R TKI RP2D TBD QD	52 MET+ ST 11 TKI-naïve NSCLC	36%	NR	NR *	Dizz 55 Lipase 33 Anaemia 29
Sym015.01 Camidge 2020 [134] NCT02648724	Mixture of Abs	Sym015 2 abs anti MET 18 mg/kg -> 12 mg/kg q2w	20 MET+ 12 METex14 8 MET AMP	5/20 (25%) 3/12 (25%) 2/8 (25%)	5.5 mos	13.3% NR *	Fat. 15.7 Oedema 8 N 6 Anorexia 6 Pruritus 6 Abd Pain 6
Phase I/II Cho 2022 [135] NCT04077099	Bp Ab	REGN5093 Bi-paratopic MET ab 2000 mg q3w	44 MET+ 11 METex14 10 METex14 tx N 35 METAMP/OE	0/11 3/10 (30%) 5/35 (12.5%)	NR NR NR	25% NR *	NR *

Abbreviations: bsAb = bispecific antibodies; ADC = antibody–drug conjugate; MET-I = MET inhibitor; Ab (s) = monoclonal antibody (ies). BpAb = biparotopic antibody Ref = reference; Pt N = patient number; ORR = overall response rate; mPFS = median progression-free survival; G3/4 R = G3/4 rate; mos = months; Int. = interruptions; Mod. = modifications; Disc. = discontinuations; EGFR = epidermal growth factor receptor; SRC = protooncogene non-receptor tyrosine kinase; CSF1R = colony-stimulating factor-1 receptor; METex14 = MET exon 14 skipping mutations; Tx naïve = treatment-naïve; MET AMP = MET gene amplification; MET OE = MET overexpressed; Sq = squamous; NR = not reported; NR * not reported in the abstract form; IRR = infusion-related reactions; SR = skin rash; DA = dermatitis acneiform; par. = paronychia; Fat. = fatigue; alb. = hypoalbuminemia; NP = neuropathy; N = nausea; Dizz = dizziness; Abd pain = abdominal pain.

Additionally, other clinical trials are ongoing with combinations of drugs to augment their efficacy (TKI plus MEK-I or ICI plus amivantamab).

## 8. Discussion

The MET pathway was discovered in 1984 [6]. Since then, several landmark studies have validated its important role in carcinogenesis and cancer progression [12,13,14]. MET alterations were then identified in several cancers, including NSCLC [12,13,15,16,17]. Subsequently, several retrospective studies have proven that patients with MET+ NSCLC have worse clinical outcomes regardless of the type of MET alteration: MET OE, MET AMP, or METex14 [18,19,20,21,22,23,24,25,26,27,28,29,30]. Of note, all these studies analyzed a mono-institutional surgical series of patients, and they were retrospective, and only two studies included all MET parameters as putative prognostic factors for survival [29,30].

Despite the growing awareness of the MET role in NSCLC, many years had to pass before the identification of the real molecular driver(s) in this disease. METex14 mutations were first described and then characterized around 2004–2006 [31,32,34]. In 2016, the first retrospective series described 28 METex14 out of 993 non-squamous NSCLC patients [36]. We learned that METex14 is extremely rare (around 3%), most of the patients have an older median age, and It is associated with an adenocarcinoma histology, even if it can also be detected in squamous and sarcomatoid carcinomas [36,37]. The most recent analysis of patients with METex14 NSCLC was presented at the last ASCO meeting involving 711 METex14 cases where 288 discrete METex14 mutations had been identified [37]. Importantly, they observed that most of the patients were ever smokers and underlined that METex14 NSCLCs are a heterogeneous disease where the incidence of other concomitant gene mutations (such as p53, POT1, TERT, among the others) varies a lot (ranging from 0.9 to 60%) because of the histology and type of METex14 mutations [37]. Due to its extreme rarity and difficulty detecting (not all mutations lead to exon 14 skipping), the ESMO guidelines recommend testing patients through an NGS panel (DNA-RNA-based should be the optimal choice) on tumour tissue or liquid biopsy [48]. The other putative driver, MET AMP, is more complex since the cut-offs chosen and validated for testing MET as prognostic factors were not helpful as predictive parameters for sensitivity to MET-Is. Currently, we know that the real MET AMP (MET/CEP7 ratio ≥ 5) is extremely rare, being less than 1% of the cases, and it is rarely associated with other concomitant gene mutations [49]. In our opinion, only these cases should be catered for clinical trials with MET-Is.

Furthermore, several MET-Is have been developed after the failure of the first approaches (onartuzumab and tivantinib) [53,54,146]. If the activity of type Ia non-selective MET inhibitor crizotinib is modest [66,67,68], the advent of more potent and selective type of Ib inhibitors, mainly capmatinib and tepotinib, allowed us to reach great results in terms of ORR and mPFS in pretreated patients (ORR 41–47%, mPFS 5.4–11 months) and especially in treatment-naïve patients (ORR 60–68%, mPFS 12.4–15.9 months) [74,87]. Moreover, their activity seems to be irrespective of age, histology, type/localization of METex14 mutations, and prior treatments [74,85]. These results led to FDA and EMA approval of capmatinib and tepotinib in patients with METex14 NSCLC irrespective of prior treatment lines in the USA and after progressing on standard treatment in Europe [48,147,148,149,150,151]. Neither capmatinib nor tepotinib have been approved for MET AMP NSCLC [48,151].

Despite the initial paramount results, most of the patients develop resistance to MET-Is through on- and/or off-target mechanisms [102]. Similar to what happens in other oncogene-addicted advanced NSCLCs, such as KRAS- or RET-driven NSCLC, specific treatments are basically lacking apart from chemotherapy and/or immunotherapy [152,153]. Therefore, new drugs are needed to overcome MET-Is resistance mediated through these mechanisms, which could help us to better manage MET+ and/or other oncogene-addicted NSCLC when they develop a MET-driven resistance [154,155]. The on-target mechanism (35% or lower) consists of acquired MET TKD mutations, and their incidence seems to be related to the type of MET inhibition. Unfortunately, only a few case reports have reported the re-sensitiveness of refractory patients changing the type of MET inhibitor [98,99,100,101], while the sole phase II trial with results is that of capmatinib in patients after progressing on crizotinib and showed very modest outcomes [104]. Moreover, the clinical trials of type II inhibitors did not address the ability to overcome MET-I resistance (Appendix A) [156,157]. The situation is more complex if we consider that beyond secondary MET mutations, 45% of cases can develop resistance to MET-Is through off-target mechanisms involving KRAS (mutations or amplifications) and gene amplification of EGFR, HER3, and BRAF [102].

There are higher expectations of relying on new drugs with different structures and functions. They consist of novel selective and potent small molecules MET-Is, bispecific/trispecific antibodies and ADCs. Thus far, no clinical trial of these drugs has been designed to investigate overcoming resistance to prior MET-Is. In fact, in all of these trials, prior MET-Is were allowed without a dedicated sub-cohort for patient refractory to prior MET-Is. So far, amivantamab is the sole drug with proven efficacy in 28 patients with METex14 NSCLC and previously pretreated with MET-Is (the ORR was 17% with an mPFS of 4.2 months) [122]. Therefore, it remains a challenge to develop new molecules and strategies to overcome MET-I resistance (on and off target).

## 9. Conclusions

METex14 can be considered a real entity in NSCLC among the other molecular drivers. Selective MET-Is (capmatinib and tepotinib) have been approved worldwide in patients with METex14 NSCLC. Given the great results obtained in non-pretreated patients, we think that the optimal place for MET-Is could have been the first line. However, in European countries, in the absence of controlled randomized trials, capmatinib and tepotinib have been approved for ICI +/− CT previously treated METex14 NSCLC patients. Great efforts in translation research and new drugs active towards MET+ NSCLC will allow us to improve these encouraging results further. The road still seems to be long but now with a favourable wind.

## Figures and Tables

**Figure 1 cancers-15-04779-f001:**
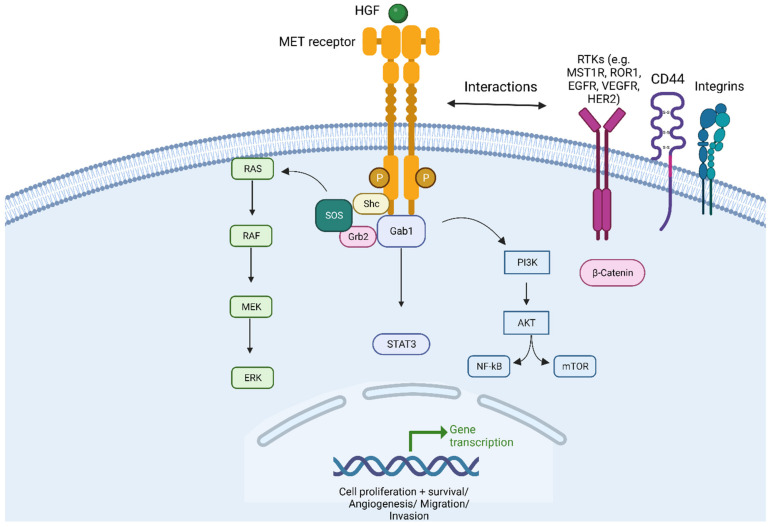
HGF/c-MET signaling pathway. Abbreviations: MET = mesenchymal–epithelial transition; HGF = Hepatocyte Growth Factor. RAS = Rat sarcoma virus proteins. BRAF = serine/threonine-protein kinase B-Raf (v-Raf murine sarcoma viral oncogene homolog B); MEK = Mitogen-activated protein kinase kinase; ERK = Extracellular signal-regulated kinases. SOS = Son of sevenless protein (also called guanine nucleotide exchange factor); Shc = Src Homology 2 Domain-Containing (adaptor protein). Grb2 = Growth factor receptor-bound protein 2; Gab1 = GRB2-associated-binding protein 1. STAT3 = signal transducer and activator of transcription 3. PIK3 = Phosphatidylinositol 3-kinases; AKT = Protein-kinase B (also called PKB); NF-kB = nuclear factor kappa-light-chain-enhancer of activated B cells; mTOR = mammalian target of rapamycin; RTKs = Receptor Tyrosine kinase pathways; MST1R = Macrophage-stimulating protein receptor 1; ROR1 = Receptor tyrosine kinase-like orphan receptor 1; EGFR = Epidermal growth factor; VEGFR = Vascular endothelial growth factor; HER2 = human epidermal growth factor receptor 2. Created with BioRender.com (accessed on 1 August 2023).

**Figure 2 cancers-15-04779-f002:**
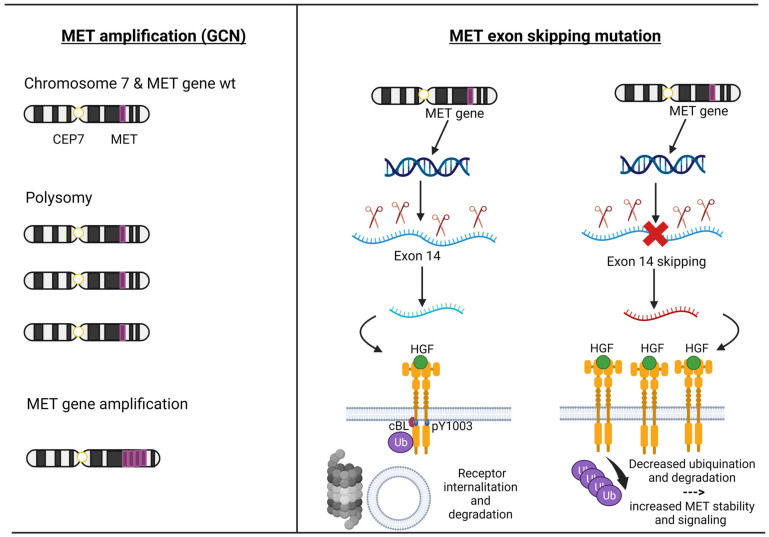
The pathogenesis of MET amplification and MET exon 14 alterations. Description: on the left, the mechanisms by which the MET gene can be amplified (polysomy versus intrachromosomal gene amplification) are depicted; the right shows the physiological mechanism of transcription and translation of the MET gene up to the degradation of the MET receptor and the pathological mechanism of the MET exon 14 skipping leading to the loss of Y1003, which decreased ubiquitination and degradation of the MET receptor. Abbreviations: MET = mesenchymal–epithelial transition; HGF = Hepatocyte Growth Factor; GCN = gene copy number; CEP7 = centromere of chromosome 7; wt = wild type. Created with BioRender.com (accessed on 18 September 2023).

**Figure 3 cancers-15-04779-f003:**
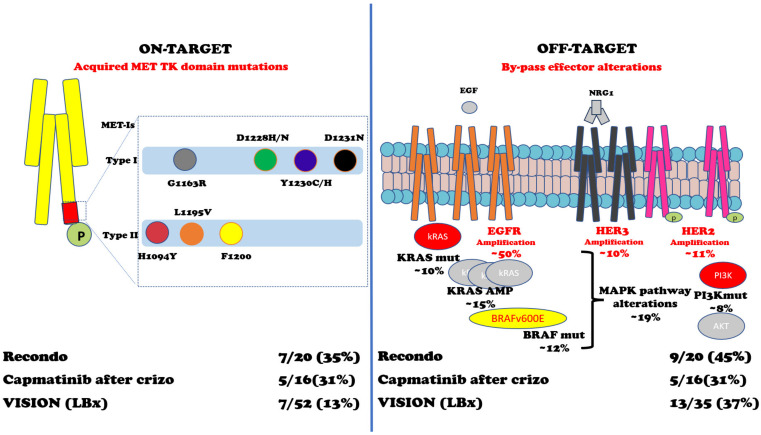
Acquired mechanisms of resistance to MET-Is. Abbreviations: MET TK domain = MET tyrosine kinase domain; MET-Is = MET inhibitors; Crizo = crizotinib; EGF = Epidermal Growth Factor; NRG1 = neuregulin 1; EGFR = EGF receptor; HER3 = human EGFR-3; HER2 = human EGFR2; kRAS = Kirsten rat sarcoma virus gene; BRAF = v-raf murine sarcoma viral oncogene homolog B1 gene; MAPK = mitogen-activated protein kinases pathway; PI3K = Phosphatidylinositol 3-kinase; AKT = Ak strain transforming kinase protein; Mut = mutations; AMP = amplification. References: Recondo G et al. [102]; Dagogo-Jack et al. [104]; Paik K et al. [95].

**Table 1 cancers-15-04779-t001:** Summary of retrospectives surgical series investigating the prognostic role of MET dysregulation.

Study Reference	Pt N	Stage	Histology	Methodology	F-Up (yrs)	Multivariate Analysis: MET Parameter Correlated with
Takanami’96 [18] [Japan]	120	I–IV	ADK	IHC	5–12	HGF/MET OE: pN and worse OS
Ichimura’96 [19] [Japan]	104	I–IV	NSCLC	WB/IHC	4	HGF/MET OE: ADK, p-Stage, worse OS
Siegfried’97 [20] [USA]	56	I–IIIA	ADK	HGF IHC	NR	HGF OE: worse DFS/OS
Tsao’98 [22] [Canada]	147	I–III	NSCLC	mRNA/IHC	NA	MET mRNA levels higher in ADK > Sq
Masuya’04 [21] [Japan]	88	I–III	NSCLC	HGF/MET IHC	4.2	HGF OE: pT, Ki-67 index, worse OS
Cheng’05 [23] [Taiwan]	45	I–IIIA	NSCLC	RT-PCR and IHC	1.9	b-MET mRNA: pN and worse DSF
Nakamura’07 [24] [Japan]	130	I–III	ADK	IHC pMET	2.7	Phospho-MET: Grade and papillary
Beau-Faller’08 [25] [France]	106	I–IV	NSCLC	GCN RT-PCR	2.2	Higher GCN: worse OS in ADK (trend)
Okuda’08 [26] [Japan]	213	I–IV	NSCLC	GCN RT-PCR	5 *	Higher GCN: worse OS (for stages II–IV)
Cappuzzo’09 [27] [Italy]	447	I–IV	NSCLC	GCN FISH	3.4	Higher GCN: p-Stage, grading, worse OS
Park’12 [28] [Korea]	380	I–IV	NSCLC	GCN FISH IHC	3.5	MET OE: worse OS in ADK (trend)
Yeung’15 [29] [Hong Kong]	154	I–IV	ADK/ADS	GCN FISH IHC RT-PCR	2.1	METex14: worse OS
Tong’16 [30] [Hong Kong]	687	I–IV	NSCLC	FISH IHC RT-PCR	3.4	METex14/High GCN: worse OS

Reference: * Not reported, prognosis was assessed as 5-year overall survival. Abbreviations: Pt N = patient number; F-up = follow-up; ADK = adenocarcinoma; NSCLC = Non-Small-Cell Lung Cancer; ADS = adenosquamous carcinoma; IHC = immunohistochemistry; HGF = Hepatocyte Growth Factor; MET = mesenchymal–epithelial transition gene; b-MET mRNA = circulating MET mRNA; phospho-MET = MET phosphorylated; RT-PCR = real-time polymerase chain reaction; MET/HGF OE = MET/HGF overexpressed; GCN = gene copy number; WB = Western Blot; METex14 = MET exon 14 skipping mutations; FISH = fluorescence in situ hybridization; DFS = disease-free survival; OS = overall survival.

**Table 2 cancers-15-04779-t002:** Summary of phase II trial of MET inhibitors in patients METex14 or MET amplified NSCLC.

	Crizotinib AcSé [66] GCN ≥ 6	Crizotinib METROS [67] MET/CEP7 ≥ 2.2	Crizotinib Profile1001 [68,69] MET/CEP7 ≥ 1.8	Capmatinib GeometryMono1 [74,75,76] any GCN		Tepotinib Vision [84,85,86,87] LB GCN ≥ 2.5		Savolitinib Phase II [93] NA	
**METAMP GCN+**	8/25	5/16	11/38	20/69		10/24		NA	
**(ORR)**	(32%)	(31%)	(29%)	(29%)		(41.7%)			
**mPFS**	3.2 mos	5.0 mos	5.1 mos	4.1 mos		4.2 mos			
**GCN High**	0/1	0/2	8/21	1 L 6/15		NR		NA	
**(ORR)**	(0)	(0)	(38%)	(40%)					
**mPFS**	NR	NR	6.7 mos	4.2 mos					
**GCN Intermediate**	5/6	5/14	2/14	5/42		NR		NA	
**(ORR)**	(83%)	(36%)	(14%)	(12%)					
**mPFS**	NR	4.4 mos	1.9 mos	2.7 mos					
**GCN Low**	2/15	NA	1/3	7/84		NR		NA	
**(ORR)**	(13%)		(33%)	(8%)					
**METex14**	10/25	2/10	21/65	1 L (28)	≥2 L (69)	1 L (95) *	≥2 L (66) *	1 L (28)	≥2 L (42)
**(ORR)**	(40%)	(20%)	(32%)	(68%)	(41%)	60%	47%	(46%)	(40%)
**mPFS**	3.6 mos	2.6 mos	7.3 mos	12.4 mos	5.4 mos	15.9 mos	12 mos	5.6 mos	6.9 mos
**METex14**	NA	NA	NA	NA		NA		10/25	
**Sarcomatoid**								(40%)	
**mPFS**								5.5 mos	

Reference: * Please note we have indicated results from confirmatory cohort C from VISION TRIAL. Abbreviations: GCN = gene copy number; LB = liquid biopsy; NA = not applicable; NR = not reported; METex14 = MET exon 14 skipping mutations; ORR = overall response rate; mPFS median progression-free survival; mos = months; 1 L = first-line; ≥2 L = second-line or further.

**Table 3 cancers-15-04779-t003:** Studies that reported resistance mechanisms to MET-Is.

Reference	Study	Drug	Type Inh.	Mechanism of Resistance
On-target				
Yao 2023 [103]	Retrosp. 32/54,752 (0.06%)	None	NA	Extremely rare incidence of MET TK domain mutations in tx-I pts
Tiedt 2011 [96]	Preclinical	NVP-BVU972 AMG458	I II	MET Y1230 MET F1200
Fujino 2019 [97]	Preclinical	MET-Is	I II	MET D1228 or Y1230 MET L1195 or F1200
Engstrom 2017 [98]	Preclinical	Type I glesatinib	I II	MET1228N or Y1230C/H Sensitive to glesatinib
Bahcall 2016 [99]	Case report	Osimertinib+ savolitinib Erlotinib+ cabozantinib	I II	EGFR+NSCLC -> MET D1228V Sensitive to cabozantinib
Heist 2016 [101]	Case report	Crizotinib	I	METex14NSCLC -> METD1228N
Ou 2017 [100]	Case report	Crizotinib	I	METex14NSCLC -> Y1230C
Recondo 2020 [102]	Retrosp. 7/15 (35%)	Crizotinib capmatinib Glesatinib	I II	G1163R, D1228H/N Y1230 L1195V H1094Y and L1195V
Yao 2023 [103]	Retrosp. 41 pts	EGFR TKI+ MET (20 pts) MET inh (21 pts)	NA	D1228N (63%) D1228H (42%) Y1230H (20%) Y1230C (15%) D1228Y (12%) L1195V (10%) D1228/M1229 (1 pt)
Dagogo-Jack 2021 [104]	Phase II 5/16 METex14 (31%)	Crizotinib	I	D1228H (2 patients) Y1230H (1 patient) D1228N/Y1230H (1 patient)
Paik 2021 [95]	Phase II 7/52 METex14 (13%)	Tepotinib	I	Y1230H/C, D1228H/N plus 3 unknown function G685E, G344R, S156L
Le 2021 [90]	Phase II 2/9 MET AMP (22%)	Tepotinib	I	D1228H/N/Y, Y1230C/H, D1231N (1) D1228N/H, Y1230H, D1231N (1)
Off-target				
Bahcall 2018 [105]	Preclinical	Crizotinib	I	KRAS gene amplification
Suzawa 2018 [106]	Retrosp. 1/113 (0.08%)	Crizotinib	I	1/113 post-Crizo KRAS mutation 4/113 pre-treatment KRAS mutations
Recondo 2020 [102]	Retrosp. 9/15 METex14 (45%)	Crizotinib Capmatinib Glesatinib	I II	KRAS gene amplification or mutations EGFR gene amplification HER3 gene amplification BRAF gene amplification
Dagogo-Jack 2021 [104]	Phase II 5/16 METex14 (31%)	Crizotinib	I	MAPK pathway alterations (3/16) ERBB pathway alterations (2/16)
Paik 2021 [95]	Phase II 13/35 METex14 (37%)	Tepotinib	I	p53/RB1 mutations (6/35), EGFR/HER2 amplifications (4/35), PI3K/RAS mutations (3/35)

Abbreviations: MET-Is = MET inhibitors; Type inh = type inhibitors; Retrosp. = retrospective; TK = tyrosine kinase; Tx naïve = treatment-naïve; pts = patients; EGFR+ = Epidermal growth factor receptor mutated; METex14 = MET exon 14 skipping mutations; MET AMP = MET amplification; Crizo = crizotinib; KRAS = Kirsten rat sarcoma virus oncogene; ERBB = EGFR family; HER2 = human epidermal growth factor receptor 2; PI3K = Phosphatidylinositol 3-kinase; MAPK = Mitogen-activated protein kinase.

**Table 4 cancers-15-04779-t004:** Retrospective studies of MET deregulated NSCLC treated with ≥ 2nd line ICI.

Reference	MET Alteration	Pt N	ORR	mPFS	Predictive Factors
Sabari 2018 [113]	METex14	24	17%	1.9 months	Neither PD-L1/mTMB
	MET AMP	0	NA	NA	NA
Dudnik 2018 [114]	METex14	14	12%	4 months	PD-L1-
	MET AMP	5	25%	4.9 months	NR
Mazieres 2019 [115]	METex14	23	NR	NR	-
	MET AMP	13	NR	NR	MET AMP-
	TOTAL	36	16%	3.4 months	Smoking status-/PD-L1-
Guiser 2020 [116]	METex14	30	36%	4.9 months	-
	MET AMP	0	NA	NA	NA

Abbreviations: Pt N = patient number; ORR = overall response rate; mPFS = median progression-free survival; METex14 = MET exon 14 skipping mutations; MET AMP = MET gene amplification; PD-L1 = Programmed Cell Death Ligand 1; mTMB = median tumour mutational burden; NA = not applicable; NR = not reported.

## Data Availability

Not applicable.

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
