# Peer review of "MET in Non-Small-Cell Lung Cancer (NSCLC): Cross ‘a Long and Winding Road’ Looking for a Target"

_cancers, 2023, doi:10.3390/cancers15194779_

Round 1

Reviewer 1 Report

The review article by Gianluca Spitaleri et al, titled MET in Non-Small-Cell Lung Cancer (NSCLC): Cross ‘a long 2 and winding road’ looking for a target, focuses on the prognostic role of MET, the summary of pivotal clinical trials of selective MET with a focus on various resistance mechanisms developed against MET Is (inhibitors).  The article is interesting in its current form but there are some suggestions to improve before publishing.

  1. Figure 2 must have a clearer font size. It is hard to read the HGF, cBl, and Y1003 in the right panel. Use the same format for Ub (ubiquitination)
  2. A graphical abstract showing normal MET and MET exon 14 skipping side by side is more relevant to understanding the consequences of MET exon 14 skipping.
  3. What is the optimal position in the treatment sequence for selective MET inhibitors for patients with NSCLC harboring MET exon 14 skipping has to be mentioned in the conclusion. 
  4. Are the acquired resistance mechanisms developed by NSCLC patients against MET epigenetical?

Author Response

Reviewer 1

The review article by Gianluca Spitaleri et al, titled MET in Non-Small-Cell Lung Cancer (NSCLC): Cross ‘a longing and winding road’ looking for a target, focuses on the prognostic role of MET, the summary of pivotal clinical trials of selective MET with a focus on various resistance mechanisms developed against MET Is (inhibitors).  The article is interesting in its current form but there are some suggestions to improve before publishing.

  • Figure 2 must have a clearer font size. It is hard to read the HGF, cBl, and Y1003 in the right panel. Use the same format for Ub (ubiquitination).

A1) We accepted this point. Now figure 2 is implemented according to the reviewer’s suggestions.

  • A graphical abstract showing normal MET and MET exon 14 skipping side by side is more relevant to understanding the consequences of MET exon 14 skipping.

A2) We thank the reviewer for this observation. We have formulated a graphical abstract as requested. Now we think that it could help more at summarizing the contents of the review.

  • What is the optimal position in the treatment sequence for selective MET inhibitors for patients with NSCLC harboring MET exon 14 skipping has to be mentioned in the conclusion.

A3) We accepted this point. Now in the conclusion section of the manuscript we have added a sentence which summarized the current place of MET-Is and what we believe should be their optimal place.

‘Given the great results obtained in non-pretreated patients, we think that the optimal place for MET-Is could have been the first line. However, in the European countries, in absence of controlled randomized trials, capmatinib and tepotinib have been approved for ICI +/- CT previously treated METex14 NSCLC patients.’  

  • Are the acquired resistance mechanisms developed by NSCLC patients against MET epigenetical?

A4) We thank the reviewer for his suggestion. Now we have added a brief sentence where we express the importance of studying genetic and epigenetic alterations to determine the causes of all resistance mechanisms to MET-Is. Unfortunately, there are currently no translational studies that have investigated epigenetic alterations that could have helped us better understand the mechanisms of resistance to inhibitory METs.

‘Moreover, there are currently no translational studies that have investigated epigenetic alterations that could have helped us better understand the mechanisms of resistance to inhibitory METs.’

Reviewer 2 Report

In this review, the Authors focus on the prognostic role of MET, summarize the pivotal clinical trials of selective MET-Is focusing on resistance mechanisms and discussing the future developments and challenges.

The topic of the review is relevant and covers a very attractive area of current interest; in fact in the literature are present a number of similar reviews.

Although it does not shine for originality and novelty, the review is containing new information, is a welcome manuscript and could represent a progress in the NSCLC field.

The manuscript is collectively well written, fluent enough and easy to follow. Tables and figures seem nice and clear. Bibliography is adequate and updated enough, but it still could be improved.

Moreover, the manuscript requires textual editing to avoid some grammatical errors and misreadings throughout the text.

Some moderate problematic issues are noted below:

The significance of some sentences is confusing and unclear.

For instance, the sentence (lines 72,73) "On the other hand, hardly it has identified them as putative molecular gene drivers and various attempts have been done to control MET+ NSCLC", could be written better.

Again, the sentence (lines 113-115) "Loss of this binding site leads to decreased ubiquination, thereby reducing MET receptor internalization and degradation, so is decreased the internalization and degradation of MET receptor [31,32], this overall leads to sustained MET activation [33]", is in part redundant and could be written better, too.

Overall, the manuscript requires textual editing to avoid some grammatical errors and misreadings throughout the text.

The authors correctly report (lines 675, 676) that "Despite the initial paramount results, most of the patients develop resistance to MET-Is through on – and/or off-target mechanisms".

Similarly, to what happens in other oncogene-addicted advanced NSCLCs (Addeo A et al., KRAS G12C Mutations in NSCLC: From Target to Resistance. Cancers, 2021; Rocco D et al., Treatment of Advanced Non-Small Cell Lung Cancer with RET Fusions: Reality and Hopes. Int J Mol Sci. 2023), specific treatments are basically lacking apart from chemotherapy and/or immunotherapy.

A brief comment in the discussion along with the relative references is strongly suggested to further support the need to develop new molecules and strategies to overcome MET-Is (as well as other selective TKIs) resistance.

The manuscript requires textual editing to avoid some grammatical errors and misreadings throughout the text.

Author Response

Reviewer 2

In this review, the Authors focus on the prognostic role of MET, summarize the pivotal clinical trials of selective MET-Is focusing on resistance mechanisms and discussing the future developments and challenges.

The topic of the review is relevant and covers a very attractive area of current interest; in fact in the literature are present a number of similar reviews.

Although it does not shine for originality and novelty, the review is containing new information, is a welcome manuscript and could represent a progress in the NSCLC field.

The manuscript is collectively well written, fluent enough and easy to follow. Tables and figures seem nice and clear. Bibliography is adequate and updated enough, but it still could be improved.

Moreover, the manuscript requires textual editing to avoid some grammatical errors and misreadings throughout the text.

  • Some moderate problematic issues are noted below: The significance of some sentences is confusing and unclear. For instance, the sentence (lines 72,73) "On the other hand, hardly it has identified them as putative molecular gene drivers and various attempts have been done to control MET+ NSCLC", could be written better.

B1) We thank the reviewer for these observations, now the English has been improved.

“On the other hand, several years had to pass before identifying the true therapeutic target among all these factors.

  • Again, the sentence (lines 113-115) "Loss of this binding site leads to decreased ubiquination, thereby reducing MET receptor internalization and degradation, so is decreased the internalization and degradation of MET receptor [31,32], this overall leads to sustained MET activation [33]", is in part redundant and could be written better, too.

B2) We thank again the reviewer for these observations, now the English has been improved.

Loss of this binding site leads to decreased ubiquination, thereby reducing MET receptor internalization and degradation resulting in increased MET levels [31-33]".

  • Overall, the manuscript requires textual editing to avoid some grammatical errors and misreadings throughout the text.

B3) We thank the reviewer for these observations, now the English has been improved.

4)         The authors correctly report (lines 675, 676) that "Despite the initial paramount results, most of the patients develop resistance to MET-Is through on – and/or off-target mechanisms".Similarly, to what happens in other oncogene-addicted advanced NSCLCs (Addeo A et al., KRAS G12C Mutations in NSCLC: From Target to Resistance. Cancers, 2021; Rocco D et al., Treatment of Advanced Non-Small Cell Lung Cancer with RET Fusions: Reality and Hopes. Int J Mol Sci. 2023), specific treatments are basically lacking apart from chemotherapy and/or immunotherapy.

B4) We thank the reviewer for his suggestion. Now we have added this comment in the discussion section and as well added the two references in the reference section.

‘Similarly, to what happens in other oncogene-addicted advanced NSCLCs such as KRAS or RET-driven NSCLC, specific treatments are basically lacking apart from chemotherapy and/or immunotherapy’.

  • A brief comment in the discussion along with the relative references is strongly suggested to further support the need to develop new molecules and strategies to overcome MET-Is (as well as other selective TKIs) resistance.

B5) We thank the reviewer for his suggestion. Now we have added a brief comment about this topic and the corresponding references.

‘Therefore, new drugs are needed to overcome MET-Is resistance mediated through these mechanisms which could help us to better manage MET+ and/or other oncogene-addicted NSCLC when they develop a MET-driven resistance’

Reviewer 3 Report

In the submitted manuscript, Spitaleri et al. provided a detailed description of aberrant c-MET signalling in NSCLCs. The authors summarized the outcomes of clinical trials in which patients were selectively treated with type Ia, Ib, and II MET inhibitors. The outcomes of clinical trials also correlated with the type of MET receptor mutation and concomitant gene mutations. Drug resistance in patients with NSCLC has been discussed and both targeted and off-target mechanisms have been considered. This review  summarizes the results of pivotal clinical trials designed for patients with dysregulated MET signalling in NSCLC.  

Author Response

Reviewer 3

In the submitted manuscript, Spitaleri et al. provided a detailed description of aberrant c-MET signalling in NSCLCs. The authors summarized the outcomes of clinical trials in which patients were selectively treated with type Ia, Ib, and II MET inhibitors. The outcomes of clinical trials also correlated with the type of MET receptor mutation and concomitant gene mutations. Drug resistance in patients with NSCLC has been discussed and both targeted and off-target mechanisms have been considered. This review  summarizes the results of pivotal clinical trials designed for patients with dysregulated MET signalling in NSCLC. 

C: We thank the reviewer for having revised our manuscript.